# Blood coagulation parameter abnormalities in hospitalized patients with confirmed COVID-19 in Ethiopia

**Shambel Araya**[1,2]\*, **Mintesnot Aragaw Mamo**[1,2], **Yakob Gebregziabher Tsegay**[3,4], **Asegdew Atlaw**[2], **Aschalew Aytenew**[2], **Abebe Hordofa**[2], **Abebe Edao Negeso**[1], **Moges Wordofa**[1], **Tirhas Niguse**[1], **Mahlet Cheru**[1], **Zemenu Tamir**[1]

**1** Department of Medical Laboratory Sciences, College of Health Sciences, Addis Ababa University, Addis Ababa, Ethiopia, **2** Department of Medical Laboratory, Millennium COVID-19 Treatment and Care Centre, St. Paual Millennium Medical College, Addis Ababa, Ethiopia, **3** Department of Medical Biotechnology, Institute of Biotechnology, University of Gondar, Gondar, Ethiopia, **4** Department of Research and Development Center, College of Health Sciences, Defense University, Addis Ababa, Ethiopia

\* shambelaraya8@gmail.com

**Data Availability Statement:** All relevant data are within the manuscript.

**Funding:** The author(s) received no specific funding for this work.

## Abstract

### Background

Coagulopathy and thromboembolic events are among the complications of Corona Virus disease 2019 (COVID-19). Abnormal coagulation parameters in COVID-19 patients are important prognostic factors of disease severity. The aim of this study was to analyze coagulation profiles of hospitalized COVID-19 patients in Addis Ababa, Ethiopia.

### Methods

This prospective cross-sectional study was conducted among 455 Covid-19 patients admitted at Millennium COVID-19 care and treatment center, Addis Ababa, Ethiopia from July 1-October 23, 2020. Prothrombin Time (PT), Activated Partial Thromboplastin Time (APTT) and International normalized ratio (INR) were determined on HUMACLOT DUE PLUS® coagulation analyzer (Wiesbaden, Germany). In all statistical analysis of results, $p < 0.05$ was defined as statistically significant.

### Result

A prolonged prothrombin time was found in 46.8% of study participants with COVID-19 and a prolonged prothrombin time and elevated INR in 53.3% of study subjects with severe and 51% of critically COVID patients. Thrombocytopenia was detected in 22.1% of COVID-19 patients. 50.5% and 51.3% of COVID-19 patients older than 55 years had thrombocytopenia and prolonged APTT respectively.

### Conclusion

In this study, prolonged prothrombin time and elevated INR were detected in more than 50% of severe and critical COVID-19 patients. Thrombocytopenia and prolonged APTT were dominant in COVID-19 patients older than 55 years. Thus, we recommend emphasis to be

**Competing interests:** The authors have declared that no competing interests exist.

given for monitoring of platelet count, PT, APTT and INR in hospitalized and admitted COVID-19 patients.

## Introduction

Coronavirus disease 2019 (COVID-19) is caused by a novel beta corona virus called severe acute respiratory syndrome coronavirus 2 (SARS-CoV-2) [1]. COVID-19 has become a pandemic that has affected the global population. As of November 8, 2020, there have been more than 49 million confirmed cases of COVID-19 and more than 1.2 million deaths, reported to World Health organization (WHO). Similarly, there have been 99,204 confirmed cases of COVID-19 with 1,518 deaths in Ethiopia [2].

The severity of COVID-19 infection ranges considerably from asymptomatic to life threatening, with lung injury being the main clinical manifestation. Most of the patients have a favorable prognosis, but some rapidly progress to severe respiratory distress syndrome, coagulation dysfunction and multiple organ failures [3, 4]. Although the pathophysiology and the underlining mechanisms of clinical manifestations remain unclear, thrombo inflammation and cytokine storm are clearly established components in Severe Acute Respiratory Distress Syndrome (SARS) of COVID-19 [5–8].

Coagulopathy and abnormal coagulation parameters were indicated among the most significant biomarkers of poor prognosis in COVID-19 patients [9–11]. A retrospective cohort study conducted in Spain Madrid demonstrated that COVID-19 non-survivors had significantly lower prothrombin time, abnormal coagulation parameters such as prolonged PT, APTT, higher D- dimer and higher fibrinogen levels compared to survivors indicating coagulation parameters could be an efficient measure for predicting the prognosis of patients with SARS COV-2 [7] and used as guiding clinical management. Similarly, Long et al. has reported that occurrence of coagulation dysfunction is more likely in severe and critically ill patients. The study also showed that D-dimer and prothrombin time could be considered as main indicators in predicting the mortality of COVID-19 patients [3]. Many studies have also demonstrated the increased occurrence of intravascular disseminated coagulopathy (DIC) in patients with COVID-19 [12, 13]. The result of blood coagulation profiles in COVID-19 patients can also guide management decisions and improve outcomes [12, 14].

Moreover, routine coagulation parameter tests results could potentially be utilized in symptomatic patients in resource limited settings with inadequate access to COVID-19 RT-PCR, as in Ethiopia, to raise suspension of this infection. However, data on coagulation profiles among Ethiopian COVID-19 patients is not readily available. Thus, the aim of this study was to determine the coagulation profile of COVID-19 patients admitted at Millennium COVID-19 treatment center, Addis Ababa, Ethiopia.

## Methods

### Ethical consideration

Ethical clearance was obtained and approved by Addis Ababa University College of Health Sciences, department of Medical Laboratory Sciences research ethics review committee (DRERC/538/20/MLS) and it was in accordance with the principles of the Helsinki II declaration. Laboratory test results were communicated to the responsible clinicians working at COVID-19 care and treatment center. Written informed consent was obtained from the study participants. All the personal identifying information obtained from the study participants were kept confidential.

## Study population

In this study, we have included 455 consecutive patients with confirmed SARS-CoV-2 infection admitted to Millennium COVID-19 treatment center, Addis Ababa, Ethiopia from July 1- October 23, 2020. The treatment center is the first referral center for COVID-19 patients in Ethiopia, since May 02, 2020. None of the study participants were receiving anticoagulant medications before blood drawing. Diagnosis of SARS-CoV-2 infection was made with real time RT-PCR.

## Sample collection and coagulation profile analysis

**Laboratory analysis.** Venous bloods were collected by professional nurses working in the COVID-19 care and treatment center: 5 mL in EDTA and 3 mL in 3.2% sodium citrated anti-coagulated tube for analysis of coagulation parameters. The samples for coagulation profile tests were collected at hospital admission. The prothrombin time (PT), activated partial pro-thrombin time (APTT), and international normalized ratio (INR) were analyzed using HUMACLOT DUE PLUS® coagulation analyzer (Wiesbaden ®, Germany). Platelet count was performed using UniCel® DxH 800 Coulter®Cellular Analysis System (Beckman Coul-ter ®, Inc. 4300 N. Harbor Blvd. Fullerton, CA 92835). The coagulation parameters were com-pared with the manufacturer cut off normal range of PT = 11.7-15 seconds, APTT = 23.8-37.9 seconds, INR = 1.0-1.2 and PLT = 159-386/µ.l. The coagulation parameters above the cut off value were considered as a prolonged and thrombocytopenia in the case of lower than cut off value for platelets. All laboratory tests and interpretation were done following the manufactur-ers' recommendation and standard operating procedures set out by the center.

## Statistical analysis

Statistical Package for the Social Sciences (SPSS) software version 25.0 (SPSS® Inc., Chicago, IL, USA) was used for statistical analysis. Chi-square test was used to determine association among categorical variables. The quantitative data were expressed as Mean ± SD. P value < 0.05 was considered as statistically significant.

# Results

## Socio-demographic and clinical characteristics of study participants

In this study, 455 patients diagnosed with COVID-19 were included. Among the study partici-pants, 289 (63.5%) were males. The study participants were between the age of 4 and 90 years with a mean of 49.9 ±18.3 years. From the total 455 study subjects, there were 297 mild cases, 90 severe cases, and 68 critical cases based on disease severity of COVID-19 (Table 1).

**Table 1. Socio-demographic characteristics of study participants.**

| Variables | | Frequency | Percent |
|---|---|---|---|
| Gender | Male | 289 | 63.5% |
| | Female | 166 | 36.5% |
| Age group | 0–18 years | 15 | 3.2% |
| | 18–35 years | 101 | 22.1% |
| | 36–55 years | 158 | 34.7% |
| | >55 years | 181 | 39.7% |
| Disease severity | Moderate | 297 | 65.2% |
| | Severe | 90 | 19.8% |
| | Critical | 68 | 15% |

**Table 2. Socio-demographic characteristics and disease severity of COVID-19 patients.**

| Variables | | Disease Severity | | | P-value |
|---|---|---|---|---|---|
| | | Moderate, n (%) | Severe, n (%) | Critical, n (%) | |
| Age (year) | 0–18, n = 15 | 10(66.7) | 4(26.7) | 1(6.7) | 0.283 |
| | 18–35, n = 101 | 65(64.35) | 22(21.78) | 14(13.8) | |
| | 36–55, n = 158 | 107(67.7) | 31(19.6) | 20(12.65) | |
| | >55, n = 181 | 115(63.5) | 33(18.2) | 33(8.2) | |
| Sex | Male, n = 289 | 187(64.7) | 56(19.3) | 46(15.9) | 0.045 |
| | Female, n = 166 | 110(66.2) | 34(20.4) | 22(13.2) | |

The median time from the disease onset to admission was 4 days (2–8 days). Severe and critical groups showed differences in sex ratio and age distribution. In severe (36.6%) and critical groups (48.5%), were elderly males of the age of >55 years old (Table 2).

## Magnitude of coagulation abnormalities

In this study, 209 COVID-19 patients (46%) showed prolonged PT and elevated INR values. Among those study participants with prolonged PT, 51.3% were above 55 years of age. Prolonged PT values were demonstrated more frequently among males (49.8%) than females (41%) and this difference was statistically significant (P = 0.045). Similarly, 51.4% and 53.3% of ICU (critical) and severe patients had prolonged PT values. Notably, prolonged APTT values were found among 43.1% of individuals, and from these 47%, 45% and 41% were among ICU (critical), severe and moderate patients, respectively. 57.2% of female patients had prolonged APTT; and 51.3% of patients aged older than 55 years had a prolonged APTT.

Thrombocytopenia was detected in 22.1% (101 out of 455) study subjects. 50.5% (50 out of 99) patients aged older than 55 years had thrombocytopenia and the occurrence was higher among male (23.8%) than female (18%) ICU patients (Table 3).

**Table 3. Result of coagulation parameters in patients with severe COVID-19 according to different variables.**

| Coagulation parameters | | Variables | | | | | | | | |
|---|---|---|---|---|---|---|---|---|---|---|
| | | Age | | | | Sex | | Disease severity | | |
| | | 0–18 n(%) | 19–35 n(%) | 36–55 n(%) | >55 n(%) | Male n(%) | Female n(%) | Moderate n(%) | Severe n(%) | Critical n(%) |
| PT | High n = 213 | 9(4.2) | 50(23.47) | 61(28.6) | 93(43.6) | 144(67.6) | 69(32.4) | 130(61) | 48(22.5) | 35(16.4) |
| | Normal n = 220 | 6(2.7) | 45(20.45) | 89(40.4) | 80(36.3) | 131(59.5) | 89(40.4) | 149(67.7) | 40(18.1) | 31(14.1) |
| | Low = 22 | 0 | 6(27.2) | 8(36.3) | 8(36.3) | 14(63.6) | 8(36.3) | 18(81) | 2(9) | 2(9) |
| APTT | High = 196 | 6(3) | 46(23.4) | 68(34.7) | 76(38.77) | 101(51.5) | 95(48.5) | 115(58.67) | 41(21) | 42(21.4) |
| | Normal n = 193 | 6(3.1) | 38(19.7) | 70(36.2) | 79(41) | 136(70.4) | 57(29.5) | 137(71) | 36(18.6) | 21(10.8) |
| | Low n = 66 | 3(4.5) | 17(25.7) | 20(30.3) | 26(39.3) | 52(78.7) | 14(21) | 45(68) | 13(19.7) | 5(7.5) |
| PLT | High n = 65 | 4(6.1) | 11(17) | 24(37) | 26(40) | 43(66) | 22(33.8) | 39(60) | 8(12.3) | 8(12.3) |
| | Normal n = 289 | 8(3) | 70(24.2) | 105(36.3) | 105(36.3) | 175(60.8) | 114(39) | 214(74) | 44(15) | 31(10.6) |
| | Low n = 101 | 3(2.9) | 20(20) | 28(27.7) | 50(49.5) | 69(69.70) | 30(30.3) | 33(32) | 38(37.6) | 30(29.7) |
| INR | High n = 210 | 9(4.2) | 50(24.7) | 60(28.5) | 91(43.3) | 141(67) | 69(32.8) | 127(60.4) | 50(23.8) | 33(15.7) |
| | Normal = 224 | 5(2.2) | 44(19.6) | 93(41.5) | 82(36.6) | 115(51) | 75(33.4) | 113(50.4) | 45(20) | 32(14.2) |
| | Low n = 21 | 1(4.7) | 7(33.3) | 5(23.8) | 8(38) | 14(66.6) | 7(33.3) | 15(71) | 3(14.5) | 3(14.5) |

PLT = platelet; PT = prothrombin time; APTT = activated partial thromboplastin time; INR = international normalized ratio.

## Discussion

The COVID-19 pandemic had a major impact on health care globally. COVID-19 has already caused >1.2 million deaths worldwide and more than 1400 in Ethiopia as of October 30,2020 according to WHO report [15]. Coagulation abnormalities are frequent in COVID-19 patients and are associated with poor prognosis and reduced survival [7]. The dysregulation of coagulation associated with hypercoagulability manifests as venous and arterial thrombosis and multi-organ dysfunction [16] which are poor prognostic markers [13, 14, 17–19]. Previous studies indicated that the coagulopathy in patients hospitalized with COVID-19 is characterized by increases in coagulation parameters such as PT, APTT and INR [20, 21].

Patients with serious infection are more likely to have COVID-19 associated coagulopathy than patients with a mild infection [21, 22]. In this recent study, prolonged PT, APTT and INR were more frequent among severe and critical COVID-19 patients. Similarly, studies also reported that thrombotic complications are common among COVID-19 patients admitted to intensive care unit (ICU) [22–24].

Treatment of the underlying condition is suggested to be paramount in coagulopathies. It is shown that bleeding is not common clinical manifestation in COVID-19 infections despite abnormal coagulation parameters [23, 24] and supportive care including blood product transfusion should be individualized in COVID -19 patients [25, 26]. Laboratory findings alone should not be taken as basis for instituting blood transfusion therapy, rather it should be reserved for those who are bleeding, requires an invasive procedure, or who are otherwise at high risk for bleeding complications [26, 27].

Published studies indicate that COVID-19 is associated with a hyper-coagulable state. Venous thromboembolism (VTE) and arterial thrombosis ranging from 15% to 30% were found in critically ill patients with COVID-19 and about 7% in those admitted to medical wards [28–30]. Abnormal thrombosis of different medical devices, deep vein thrombosis and multiple thrombi in the vessels of the lungs, kidneys and other organs at autopsy of patients who died of Covid-19 have been reported serving as the impetus behind guidelines [9, 29] which support the use of therapeutic doses of heparin or low-molecular-weight heparin instead of prophylactic doses in critically ill COVID-19 patients [12, 26, 31]. In the current study, thrombocytopenia was observed more frequently among males (23.8%) than females (19.8%) and older people (27.6%). Severe (42.68%) and critical (42%) patients also more frequently had thrombocytopenia and this was in line with studies conducted in different countries [20, 22, 32, 33]. Thrombocytopenia, defined as platelet count less than $100\times10^9$ cells/L were independently associated with COVID-19 severity [34]. Studies suggest that routine coagulation test results are markers of disease severity and assist in management decision. In critically ill patients, thrombocytopenia correlates with multi-organ failure and death, and a decline in platelet count, even in the absence of overt thrombocytopenia, portends a worse outcome [9, 12, 13]. In patients who are not bleeding, there is no evidence that correction of laboratory parameters with blood products improves outcomes. Replacement might worsen disseminated thrombosis and further deplete scarce blood products [28, 35].

Many studies reported that coagulopathy associated with COVID-19 is characterized by thrombocytopenia, prolongation of the prothrombin time, high levels of D-dimer, and elevated levels of fibrinogen, factor VIII, and von Willebrand factor [3, 11, 16]. Published studies indicate that COVID-19-associated coagulopathy is characterized by a decreased platelet count [9, 36–38] and a cytokine storm with an extreme hyper-coagulable state. Even though the reason for this life-threatening condition is not known, this might be due to an uncontrolled hyper-inflammatory response without previous immunity [39, 40].

## Conclusion

In this study, prolonged prothrombin time and high INR were found among severe and critical COVID-19 patients. Thrombocytopenia and prolonged clotting time assay were dominant in COVID-19 patients older than 55 years. Thus, we recommend emphasis to be given for monitoring of platelet count, PT, APTT and INR in hospitalized COVID-19 patients management.

## Author Contributions

**Conceptualization:** Shambel Araya, Mintesnot Aragaw Mamo, Yakob Gebregziabher Tsegay, Asegdew Atlaw, Aschalew Aytenew, Abebe Hordofa, Abebe Edao Negeso, Moges Wordofa, Tirhas Niguse.

**Data curation:** Shambel Araya, Mintesnot Aragaw Mamo, Yakob Gebregziabher Tsegay, Asegdew Atlaw, Aschalew Aytenew, Abebe Hordofa, Abebe Edao Negeso, Moges Wordofa, Tirhas Niguse, Mahlet Cheru, Zemenu Tamir.

**Formal analysis:** Shambel Araya, Mintesnot Aragaw Mamo, Yakob Gebregziabher Tsegay, Asegdew Atlaw, Aschalew Aytenew, Abebe Hordofa, Abebe Edao Negeso, Moges Wordofa, Tirhas Niguse, Mahlet Cheru, Zemenu Tamir.

**Funding acquisition:** Shambel Araya, Yakob Gebregziabher Tsegay, Asegdew Atlaw, Aschalew Aytenew, Abebe Hordofa, Abebe Edao Negeso, Moges Wordofa, Tirhas Niguse, Mahlet Cheru, Zemenu Tamir.

**Investigation:** Shambel Araya, Mintesnot Aragaw Mamo, Yakob Gebregziabher Tsegay, Asegdew Atlaw, Abebe Hordofa, Abebe Edao Negeso, Moges Wordofa, Tirhas Niguse, Mahlet Cheru, Zemenu Tamir.

**Methodology:** Shambel Araya, Mintesnot Aragaw Mamo, Yakob Gebregziabher Tsegay, Asegdew Atlaw, Aschalew Aytenew, Abebe Hordofa, Abebe Edao Negeso, Moges Wordofa, Tirhas Niguse, Mahlet Cheru, Zemenu Tamir.

**Project administration:** Shambel Araya, Abebe Edao Negeso, Tirhas Niguse.

**Resources:** Shambel Araya, Mintesnot Aragaw Mamo, Yakob Gebregziabher Tsegay, Asegdew Atlaw, Aschalew Aytenew, Abebe Hordofa, Abebe Edao Negeso, Moges Wordofa, Tirhas Niguse, Mahlet Cheru, Zemenu Tamir.

**Software:** Shambel Araya, Mintesnot Aragaw Mamo, Yakob Gebregziabher Tsegay, Asegdew Atlaw, Aschalew Aytenew, Abebe Hordofa, Abebe Edao Negeso, Moges Wordofa, Tirhas Niguse, Mahlet Cheru, Zemenu Tamir.

**Supervision:** Shambel Araya, Yakob Gebregziabher Tsegay, Asegdew Atlaw, Aschalew Aytenew, Abebe Hordofa, Tirhas Niguse, Zemenu Tamir.

**Validation:** Shambel Araya, Mintesnot Aragaw Mamo, Yakob Gebregziabher Tsegay, Asegdew Atlaw, Abebe Hordofa, Abebe Edao Negeso, Moges Wordofa, Tirhas Niguse, Mahlet Cheru, Zemenu Tamir.

**Visualization:** Shambel Araya, Mintesnot Aragaw Mamo, Yakob Gebregziabher Tsegay, Asegdew Atlaw, Aschalew Aytenew, Abebe Hordofa, Abebe Edao Negeso, Moges Wordofa, Tirhas Niguse, Mahlet Cheru, Zemenu Tamir.

**Writing – original draft:** Shambel Araya, Aschalew Aytenew, Abebe Edao Negeso, Tirhas Niguse, Zemenu Tamir.

**Writing – review & editing:** Shambel Araya, Mintesnot Aragaw Mamo, Yakob Gebregziabher Tsegay, Asegdew Atlaw, Aschalew Aytenew, Abebe Hordofa, Abebe Edao Negeso, Moges Wordofa, Tirhas Niguse, Mahlet Cheru, Zemenu Tamir.

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
