## [Decision Letter · Decision Letter 0]

18 Dec 2020

PONE-D-20-35898

Blood coagulation parameter abnormalities among patients with confirmed COVID-19 in Ethiopia

PLOS ONE

Dear Dr. Araya,

Thank you for submitting your manuscript to PLOS ONE. After careful consideration, we feel that it has merit but does not fully meet PLOS ONE’s publication criteria as it currently stands. Therefore, we invite you to submit a revised version of the manuscript that addresses the points raised during the review process.

This is an important and timely paper to consider the abnormalities of haematological parameters in an African context and in COVID-19 disease. There are, however, unfortunately significant numbers of typographical errors and word omissions which make the sense of the paper difficult to follow in some cases. For example, the authors refer to a "prolonged" but do not indicate which parameter. There are, in addition, a number of references which need to be included (especially in the discussion).

We look forward to receiving your revised manuscript.

Kind regards,

Elizabeth S. Mayne, M.D.

Academic Editor

PLOS ONE

Additional Editor Comments:

This is an important and timely paper to consider the abnormalities of haematological parameters in an African context and in COVID-19 disease. There are, however, unfortunately significant numbers of typographical errors and word omissions which make the sense of the paper difficult to follow in some cases. For example, the authors refer to a "prolonged" but do not indicate which parameter. There are, in addition, a number of references which need to be included (especially in the discussion).

Journal Requirements:

2. Thank you for submitting the above manuscript to PLOS ONE. During our internal evaluation of the manuscript, we found significant text overlap in the Discussion, Abstract, and other sections, between your submission and the following previously published works:

- https://applications.emro.who.int/emhj/v26/09/1020-3397-2020-2609-999-1004-eng.pdf

- https://www.hematology.org/COVID-19/COVID-19-and-coagulopathy

- https://pubmed.ncbi.nlm.nih.gov/32702124/

Please revise the manuscript to rephrase the duplicated text, cite your sources, and provide details as to how the current manuscript advances on previous work. Please note that further consideration is dependent on the submission of a manuscript that addresses these concerns about the overlap in text with published work.

3. Please state whether you validated the questionnaire prior to testing on study participants. Please provide details regarding the validation group within the methods section.

6. Please amend the manuscript submission data (via Edit Submission) to include author MogesWordofa.

7. We note you have included a table to which you do not refer in the text of your manuscript. Please ensure that you refer to Table 1 in your text; if accepted, production will need this reference to link the reader to the Table.

Reviewers' comments:

Reviewer's Responses to Questions

**Comments to the Author**

1. Is the manuscript technically sound, and do the data support the conclusions?

Reviewer #1: Yes

Reviewer #2: Yes

2. Has the statistical analysis been performed appropriately and rigorously? 

Reviewer #1: Yes

Reviewer #2: Yes

3. Have the authors made all data underlying the findings in their manuscript fully available?

Reviewer #1: Yes

Reviewer #2: Yes

4. Is the manuscript presented in an intelligible fashion and written in standard English?

Reviewer #1: No

Reviewer #2: Yes

5. Review Comments to the Author

Reviewer #1: This study is a simple evaluation of coagulation parameters that occur durning covid. The authors detected an increase in the prothrombin time and a decrease in the platelet count which correlated with the severity of infection. It is interesting and is worthy of publication

Reviewer #2: The study is relevant in the African context and will assist in the management of COVID patients in this often under resourced environment. The article however requires review of the English grammar.

6. PLOS authors have the option to publish the peer review history of their article (what does this mean?). If published, this will include your full peer review and any attached files.

Reviewer #1: No

Reviewer #2: No

---

## [Author Response · Author response to Decision Letter 0]

11 Jan 2021

Cover letter

Shambel Araya

Addis Ababa University

Addis Ababa, Ethiopia 

Email: shambelaraya8@gmail.com

Date: January 1, 2021

To: PLOS ONE Journal 

Dear Editorial: 

We are glad to write this response that our paper entitled “Blood coagulation parameter abnormalities among patients with confirmed COVID-19 in Ethiopia” (Submission ID: PONE-D-20-35898) has been requested to review for publication in PLOS ONE journal. We are pleased to have an opportunity to make our paper revised and we have greatly appreciated the reviewers’ and editor’s comments and suggestions were very helpful overall. In revising the paper, we have carefully considered reviewers’ and editor’s comments and suggestions on our revised submission. As instructed, we have attempted to succinctly explain changes made in reaction to all comments and reply to each comment in point-by-point fashion as follows: 

Response to Editor’s comments 

Additional Editor comments

Comment. “This is an important and timely paper to consider the abnormalities of haematological parameters in an African context and in COVID-19 disease. There are, however, unfortunately significant numbers of typographical errors and word omissions which make the sense of the paper difficult to follow in some cases. For example, the authors refer to a "prolonged" but do not indicate which parameter. There are, in addition, a number of references which need to be included (especially in the discussion).”

Response: As suggested by the editor we thoroughly went through the manuscript and revised the typographical, grammatical, editorial, and word omissions. 

Journal requirements

Comment #1. “Please ensure that your manuscript meets PLOS ONE's style requirements, including those for file naming.”

Response #1: we strictly followed PLOS ONE's style requirements during preparation and revision of our manuscript as suggested by the editor. 

Comment #2. “Thank you for submitting the above manuscript to PLOS ONE. During our internal evaluation of the manuscript, we found significant text overlap in the Discussion, Abstract, and other sections, between your submission and the following previously published works:

- https://applications.emro.who.int/emhj/v26/09/1020-3397-2020-2609-999-1004-eng.pdf

- https://www.hematology.org/COVID-19/COVID-19-and-coagulopathy

- https://pubmed.ncbi.nlm.nih.gov/32702124/

Please revise the manuscript to rephrase the duplicated text, cite your sources, and provide details as to how the current manuscript advances on previous work. Please note that further consideration is dependent on the submission of a manuscript that addresses these concerns about the overlap in text with published work.”

Response #2. As per the suggestion of the editor, we have revised different sections of manuscript to reduce text overlap with the mentioned previous studies and cite the utilized sources as follows: 

• Abstract, Background section: ‘Infection with corona virus disease 2019 (COVID-19) could be complicated with coagulopathy and high risk of thromboembolic events.’ Revised as “Coagulopathy and thromboembolic events are among the complications of Corona Virus disease 2019 (COVID-19).” 

• Abstract, conclusion section: ‘We found an abnormal pattern of coagulation parameters and association of advanced age and co-morbidities with a high rate of mortality in severe COVID-19 patients, which should be taken into consideration in their hospital management. 

Revised as “In this study, prolonged prothrombin time and high INR were found among severe and critical COVID-19 patients. Thrombocytopenia and prolonged APTT were dominant in COVID-19 patients older than 55 years. Thus, we recommend emphasis to be given for monitoring of platelet count, PT, APTT and INR in hospitalized COVID-19 patients management.”

• Discussion part first paragraph: ‘COVID-19, which is caused by SARS-CoV-2, has spread across the globe. Although most patients recover within 1 to 3 weeks, COVID-19 has already caused >1.2 million deaths worldwide and more than 1400 in Ethiopia as of October 30,2020 according to WHO report (15). Dysregulation of coagulation produces a coagulopathy associated with hyper coagulability as evidenced by venous and arterial thrombosis and multi-organ dysfunction. Up to 20% of affected patients require hospitalization, and the mortality rate in such patients is high (16, 17). Coagulopathy is one of the most significant prognostic factors in patients with COVID-19 and is associated with increased mortality and admission to critical care (14, 18). Most commonly observed coagulopathy in patients hospitalized with COVID-19 (COVID-19-associated coagulopathy) is characterized by increased coagulation parameters like PT, APTT and INR levels (19, 20).’

 Revised as “The COVID-19 pandemic has brought major impact on health care globally. It has already caused >1.2 million deaths worldwide and more than 1400 in Ethiopia as of October 30,2020 according to WHO report(15). Coagulation abnormalities are indicated as frequent findings in COVID-19 patients and associated with poor prognosis and survival(7). Similarly, it is also indicated that coagulopathy which is resulted due to dysregulation of coagulation and associated with hypercoagulability as evidenced by venous and arterial thrombosis and multiorgan dysfunction(16); is one of the most significant prognostic factors in patients with COVID-19 and associated with increased hospitalization, admission to critical care, and mortality(14, 17-19). Previous studies indicated that coagulopathy in patients hospitalized with COVID-19 is characterized by increase in coagulation parameters such as PT, APTT and INR levels(20, 21).”

• Discussion part 3rd Paragraph: “As for all coagulopathies, treatment of the underlying condition is paramount. Experience to date suggests that COVID-19 infection infrequently leads to bleeding despite abnormal coagulation parameters (23,24). Supportive care including blood product transfusion should be individualized (24, 25). Blood component therapy should not be instituted on the basis of laboratory results alone, but reserved for those who are bleeding requires an invasive procedure, or who are otherwise at high risk for bleeding complications (25).”

Revised as “Treatment of the underlying condition is suggested to be paramount in coagulopathies. It is shown that bleeding is not common clinical manifestation in COVID-19 infections despite abnormal coagulation parameters (23,24). Along these, it is suggested that supportive care including blood product transfusion should be individualized in COVID -19 patients (25, 26). Laboratory findings alone should not be taken as basis for instituting blood transfusion therapy, rather it should be reserved for those who are bleeding, require an invasive procedure, or who are otherwise at high risk for bleeding complications(26, 27)”

Discussion part 4th Paragraph: “Considerable evidence indicates that COVID-19 is associated with a hyper-coagulable state. Thus, despite anticoagulant thrombo-prophylaxis, different studies have reported that rates of venous thromboembolism (VTE) and arterial thrombosis ranging from 15% to 30% in critically ill patients with COVID-19 and ∼7% in those admitted to medical wards (26-28). Clotting of access catheters, dialysis membranes, and extracorporeal circuits has also been reported. Furthermore, in patients dying from COVID-19, autopsy studies reveal unsuspected deep vein thrombosis and multiple thrombi in the vessels of the lungs, kidneys, and other organs (9, 27). These findings have prompted some clinicians to use treatment doses of heparin or low- molecular-weight heparin instead of prophylactic doses in critically ill COVID-19 patients (12,25, 29).

Revised as “Evidences indicate that COVID-19 is associated with a hyper-coagulable state. Venous thromboembolism (VTE) and arterial thrombosis ranging from 15% to 30% were found in critically ill patients with COVID-19 and about 7% in those admitted to medical wards (28-30). Clotting is reported from different medical devices used, deep vein thrombosis and multiple thrombi in the vessels of the lungs, kidneys and other organs from autopsy of patients died of Covid-19 (9, 29). These indicate clinicians to use therapeutic doses of heparin or low-molecular-weight heparin instead of prophylactic doses in critically ill COVID-19 patients (12, 26, 31).

• Conclusion part: “We recommend monitoring platelet count, PT, APTT and INR. Worsening of these parameters indicates progressive severity of COVID-19 infection and predicts that more aggressive critical care will be needed; experimental therapies for COVID-19 infection might be considered in this setting. Improvement of coagulation parameters along with improving clinical condition provides confidence that stepping down of aggressive treatment may be appropriate.”

Revised as “In this study, prolonged prothrombin time and high INR were found among severe and critical COVID-19 patients. Thrombocytopenia and prolonged APTT were dominant in COVID-19 patients older than 55 years. Thus, we recommend emphasis to be given for monitoring of platelet count, PT, APTT and INR in hospitalized COVID-19 patients management. 

Comment #3. “Please state whether you validated the questionnaire prior to testing on study participants. Please provide details regarding the validation group within the methods section.”

Response #3. Not applicable. We did not used questionnaire in this study. 

Comment #4. We note that you have indicated that data from this study are available upon request. PLOS only allows data to be available upon request if there are legal or ethical restrictions on sharing data publicly. For information on unacceptable data access restrictions, please see http://journals.plos.org/plosone/s/data-availability#loc-unacceptable-data-access-restrictions.

Response #4. All the available data were included in the manuscript. 

Comment #5. ” PLOS requires an ORCID iD for the corresponding author in Editorial Manager on papers submitted after December 6th, 2016. Please ensure that you have an ORCID iD and that it is validated in Editorial Manager. To do this, go to ‘Update my Information’ (in the upper left-hand corner of the main menu), and click on the Fetch/Validate link next to the ORCID field. This will take you to the ORCID site and allow you to create a new iD or authenticate a pre-existing iD in Editorial Manager. Please see the following video for instructions on linking an ORCID iD to your Editorial Manager account: https://www.youtube.com/watch?v=_xcclfuvtxQ”

Response #5. “Already linked”

Comment #6. “Please amend the manuscript submission data (via Edit Submission) to include author Moges Wordofa.”

Response #6. Comment accepted and author included

Comment #7. “We note you have included a table to which you do not refer in the text of your manuscript. Please ensure that you refer to Table 1 in your text; if accepted, production will need this reference to link the reader to the Table.”

Response #7. Comment accepted and corrected accordingly. 

Response to comments and suggestions inserted in the PDF format Manuscript 

Comment # 1: in the abstract section. A prolonged (insert analytical parameter that was prolonged) was present in 48.8% of study subjects with COVID-19? 

Response 1: Parameter inserted and and revised as “A prolonged prothrombin time was found among 46.8% of study subjects with COVID-19”

Comment # 2: In the abstract section. The comment to remove the interpretation and conclusion.

Response #2. Comment accepted and modified accordingly. 

Comment #3. On Title. “Blood coagulation parameter abnormalities among in hospitalized patients with confirmed COVID-19 in Ethiopia”

Response #3: Accepted and modified as: “Blood coagulation parameter abnormalities in hospitalized patients with confirmed COVID-19 in Ethiopia”

Comment#4: in the abstract, method part: ‘…were estimated by auto analyzer.” ‘Which parameter?’.

Response to #5. Although the comment was to indicate which parameter, since the parameter are already mentioned we perceived it as to mean which auto analyzer and inserted the specific analyzer used as_”…..HUMACLOT DUE PLUS coagulation analyzer (Wiesbaden, Germany)” 

Comment #6. In the introduction part. “ ….WHO. Write out abbreviation in full the first time it is used”.

Response #6. Corrected as suggested. …..World Health Organization(WHO). 

Comment #7. In the laboratory analysis part, “…HUMACLOT DUE PLUS.. Insert trademark symbol e.g. R or TM.” 

Response #7. Accepted and modified as “…HUMACLOT DUE PLUS coagulation analyzer (Wiesbaden, Germany)

Comment #8. In the laboratory analysis part, “…Beckman coulter DxH 600 automated hematology analyzer.. Insert trademark symbol e.g. R or TM.” 

Response #8. Accepted and modified as “…Beckman coulter DxH 600 automated hematology analyzer

Comment #9. “Thrombocytopenia and abnormal coagulation parameters (PT, APTT and INR) could be considered as important indicators of COVID-19 disease severity. This statement belongs in the discussion section of the article.”

Response # 9: Accepted and removed as suggested. 

Comment #10. In discussion part. ‘Thrombocytopenia, defined as platelet count less than 100×10⁹ cells per L were independently associated with severity. Insert reference.’

Response #10: Comment accepted; reference inserted as suggested. “Thrombocytopenia, defined as platelet count less than 100×10⁹cells/L were independently associated with COVID-19 severity(34)”

Comment 11. Discussion Part. “As many studies reported that the coagulopathy associated with COVID-19 is characterized by thrombocytopenia, prolongation of the prothrombin time, high levels of D-dimer, and elevated levels of fibrinogen, factor VIII, and von Willebrand factor. Insert reference” 

Response #11. Comment accepted; reference inserted as suggested. “Many studies reported that coagulopathy associated with COVID-19 is characterized by thrombocytopenia, prolongation of the prothrombin time, high levels of D-dimer, and elevated levels of fibrinogen, factor VIII, and von Willebrand factor(3, 11, 16).” 

Comment #12. Conclusion part. “We recommend monitoring platelet count, PT, APTT and INR. ...in patients hospitalized COVID-19 patients.

Response #12. Comment accepted and modified as “We recommend monitoring platelet count, PT, APTT and INR in hospitalized COVID-19 patients.”

Looking forward to hearing from you. Thank you again for your consideration! 

Sincerely, 

Shambel Araya (BSc, MSc)

Corresponding author

---

## [Decision Letter · Decision Letter 1]

31 Mar 2021

PONE-D-20-35898R1

Blood coagulation parameter abnormalities in hospitalized patients with confirmed COVID-19 in Ethiopia

PLOS ONE

Dear Dr. Araya,

Thank you for submitting your manuscript to PLOS ONE. After careful consideration, we feel that it has merit but does not fully meet PLOS ONE’s publication criteria as it currently stands. Therefore, we invite you to submit a revised version of the manuscript that addresses the points raised during the review process.

Although both reviewers were happy that their concerns were addressed, there are some minor predominantly typographical and grammatical errors.

Please review attached reviews.

We look forward to receiving your revised manuscript.

Kind regards,

Elizabeth S. Mayne, M.D.

Academic Editor

PLOS ONE

Journal Requirements:

Reviewers' comments:

Reviewer's Responses to Questions

**Comments to the Author**

1. If the authors have adequately addressed your comments raised in a previous round of review and you feel that this manuscript is now acceptable for publication, you may indicate that here to bypass the “Comments to the Author” section, enter your conflict of interest statement in the “Confidential to Editor” section, and submit your "Accept" recommendation.

Reviewer #2: All comments have been addressed

2. Is the manuscript technically sound, and do the data support the conclusions?

Reviewer #2: Yes

3. Has the statistical analysis been performed appropriately and rigorously? 

Reviewer #2: Yes

4. Have the authors made all data underlying the findings in their manuscript fully available?

Reviewer #2: Yes

5. Is the manuscript presented in an intelligible fashion and written in standard English?

Reviewer #2: Yes

6. Review Comments to the Author

Reviewer #2: The study adds to local knowledge of the clinical manifestations and outcomes in patients with COVID and has the potential to assist with clinical decisions. Various final changes have been recommended as indicated on the uploaded edited article.

7. PLOS authors have the option to publish the peer review history of their article (what does this mean?). If published, this will include your full peer review and any attached files.

Reviewer #2: No

---

## [Author Response · Author response to Decision Letter 1]

2 Apr 2021

Cover letter

Shambel Araya

Addis Ababa University

Addis Ababa, Ethiopia 

Email: shambelaraya8@gmail.com

Date: April, 2021

To: PLOS ONE Journal 

Dear Editorial: 

We are glad to write this response that our research article entitled “Blood coagulation parameter abnormalities among patients with confirmed COVID-19 in Ethiopia” (Submission ID: PONE-D-20-35898R1) has been requested to review for publication in PLOS ONE journal. We are pleased to have an opportunity to make our paper revised and we have greatly appreciated the reviewers’ and editor’s comments and suggestions were very helpful overall. In revising the paper, we have carefully considered reviewers’ and editor’s comments and suggestions on our revised submission. As instructed, we have attempted to succinctly explain changes made in reaction to all comments and reply to each comment in point-by-point fashion as follows: 

Response to comments and suggestions inserted in the PDF format of the Manuscript 

Comment 1: Abstract, methods section: “...Estimated by...” was replaced by “...determined on...” as per your suggestion in line 34

Comment 2: Abstract, Result section: “among” is replaced with ‘in’ in line 36

Comment 3: Abstract, Result section: “…Prolonged prothrombin time and high INR were seen among 53.3% severe and 51% critical 38 patients with COVID-19 manifestation…. Thrombocytopenia was detected in 22.1% of COVID-19 39 patients” has replaced with… and a prolonged prothrombin time and elevated INR with 53.3% of study subjects with severe and 51 % of critically COVID patients…” in line 37-39

Comment 4: abstract, conclusion section. In this study, prolonged prothrombin time and high INR were found among severe and critical COVID-19 patients.

Revised: In this study, prolonged prothrombin time and high INR were detected in 50% of severe and critical COVID-19 patients

Comment: abstract, conclusion section. Delete management

Response: Accepted and deleted in line 45

Comment 6: Introduction section. Replace showed with demonstrated that

Response: Accepted and replaced

Comment 7: …Introduction section. coagulation parameters could be an efficient measure for improving the clinical management and predicting the prognosis of patients with SARS COV-2(7)…

Revised: …coagulation parameters could be an efficient measure for predicting the prognosis of patients with SARS COV-2(7) and guiding management…

Comment 8: Introduction section: Accepted and modified as follow

“Different studies also support that COVID-19 patients are at high risk of developing disseminated intravascular coagulation (12, 13). It is also indicated that comparison of reports from various populations related to the clinical course, outcome of COVID-19 and blood coagulation profile in these patients are necessary to help the management and treatment of the disease (12, 14). Moreover, this routine coagulation parameter tests could be used as potential indicators for COVID-19 in individuals having typical clinical manifestations that would be inputs for prompt patient management especially in resource limited settings where the high-tech gold standard RT-PCR is not widely available, like Ethiopia”

Revised: Several studies have also demonstrated the increased occurrence of intravascular disseminated coagulopathy (DIC) in patients with COVID-19 (12, 13). The result of blood coagulation parameters in COVID-19 can also guide management decisions and improve outcomes (12, 14).

Moreover, routine coagulation parameter tests results could potentially be utilized in symptomatic patients in resource limited settings with inadequate access to COVID-19 RT-PCR, as in Ethiopia, to raise suspension of this infection. However, data on coagulation profiles among Ethiopian COVID-19 patients is not readily available, like Ethiopia.

Comment8: Introduction section Replace “…find out…” with “...Determine…”

Response: accepted and replaced as per your suggestion

Comment 9: Delete “and” in line 88

Line 92: Replace “…of…” with “…for…”

Line 94: Replace “…Was taking…” with “…were receiving…”

Line 95: Change “According to” in to “…with…”

Response: Accepted and modified accordingly

Comment 10: Methods, laboratory section.

Eight milliliters of venous blood were collected by professional nurses working in the treatment center: five milliliters in EDTA for platelet count, three milliliters in 3.2% sodium citrated 

Revised: Venous bloods were collected by professional nurses working in the treatment center: 5 mL in EDTA and 3 mL in 3.2% sodium citrated

Comment 11: Insert trade mark for each instruments and software’s

Response: accepted and inserted as follow in line 99-102

Comment12: remove Addis Ababa, Ethiopia, 2020 from table 1

Response: Deleted as per your suggestion

Comment 13: Result section line 120-122 “Severe and critical groups showed differences in sex ratio and age distribution. In severe and critical groups, majority were males and elderly (Table 2).”

Revised: Severe and critical groups showed differences in sex ratio and age distribution. In severe (36.6%) and critical groups (48.5%), were elderly males of the age of >55 years old. y (Table 2).

Comment 14: result section table 3: remove “…admitted to Millennium COVID-19 treatment center Addis Ababa, Ethiopia…”

Response: accepted and removed table heading

Comment 15: Result section line 127-129, rewrite “Prolonged PT value was found among males (49.8%) than females (41%) and it has a significant association with gender (P = 0.045).”

Revised: accepted and modified as “Prolonged PT values were demonstrated more frequently among males (49.8%) than females (41%) and this difference was significantly different (P = 0.045).”

Comment 16: replace study subjects with individuals

Response accepted and replaced in line 130

Comment 16 result section line 135-136: rearrange “…patients aged older than 55 years had thrombocytopenia. Thrombocytopenia was higher among…” in to

Revised: modified as “…patients aged older than 55 years had thrombocytopenia and the occurrence was higher among…”

Comment 17: insert “…Result of …” in table 3 heading

Response: accepted and modified as follow “Result of coagulation parameters in patients with severe COVID-19 according to different variables”

Comment 18: Discussion section: Replace “…had brought…” with “...had a…” 

Delete indicated as

Remove findings 

Comment 19: Rearrange the paragraph as below for “Coagulation abnormalities are indicated as frequent findings in COVID-19 patients and associated with poor prognosis and survival(7). Similarly, it is also indicated that coagulopathy which is resulted due to dysregulation of coagulation and associated with hypercoagulability as evidenced by venous and arterial thrombosis and multiorgan dysfunction (16); is one of the most significant prognostic factors in patients with COVID-19 and is associated with increased hospitalization, admission to critical care, and mortality”

Response: accepted

Revised: Coagulation abnormalities are frequent in COVID-19 patients and are associated with poor prognosis and reduced survival(7). Dysregulation of coagulation and associated with hypercoagulability in patients with COVID manifest as venous and arterial thrombosis and multiorgan dysfunction (16); which are poor prognostic markers resulting in increased mortality and hospitalization and ICU admission

Comment 20: discussion section 

Line 151: remove “…levels...”

Line 153 insert “…more frequent…”

Line 153: remove “…than moderate ones…”

Line 159: replace “… alongside these, it is suggested that…” with “and”

Line 164: replace “…Evidence…” with “…published studies…”

Response: Accepted and modified as per your suggestion

Line 167: replace “…clotting is reported from…” with “…Abnormal thrombosis of…”

Line 167-170: rewrite it as follow for “…Clotting is reported from different medical devices used, deep vein thrombosis and multiple thrombi in the vessels of the lungs, kidneys and other organs from autopsy of patients died of Covid-19 (9, 29). These indicate clinicians to use…”

Response: Accepted and rephrase as follow

Revised: “…Abnormal thrombosis of different medical devices, deep vein thrombosis and multiple thrombi in the vessels of the lungs, kidneys and other organs at autopsy of patients who died of Covid-19 have been reported serving as the impetus behind guidelines (9, 29)which support the use…”

Line 173-175: rewrite it as follow for “Studies across suggested that routine coagulation tests can be considered as a significant marker to help clinicians assess prognosis and severity of patients with COVID-19…”

Response: Accepted and rephrase as follow

Revised: (34). “...Studies suggest that routine coagulation test results are markers of disease severity and assist in management decision…”

Line 184: Insert “… in critically ill non-COVID patients…”

Response: accepted

Line 186-189: rephrase the sentence “COVID-19-associated coagulopathy is however unique with a much-decreased platelet count (9, 36-38)”

Revised: Published studies support that COVID-19-associated coagulopathy is characterized by a decreased platelet count (9, 36-38).

Dear all, we are very grateful for your valuable comments and for your time

Looking forward to hearing from you. Thank you again for your consideration! 

Sincerely, 

Shambel Araya (BSc, MSc, PhD fellow)

---

## [Decision Letter · Decision Letter 2]

26 Apr 2021

PONE-D-20-35898R2

Blood coagulation parameter abnormalities in hospitalized patients with confirmed COVID-19 in Ethiopia

PLOS ONE

Dear Dr. Araya,

Thank you for submitting your manuscript to PLOS ONE. After careful consideration, we feel that it has merit but does not fully meet PLOS ONE’s publication criteria as it currently stands. Therefore, we invite you to submit a revised version of the manuscript that addresses the points raised during the review process.

The reviewer felt that the manuscript was substantially improved but there are still a number of grammatical and typographical errors that should be corrected. Again, it is recommended that the manuscript is reviewed by a native English language speaker.

We look forward to receiving your revised manuscript.

Kind regards,

Elizabeth S. Mayne, M.D.

Academic Editor

PLOS ONE

Journal Requirements:

Reviewers' comments:

Reviewer's Responses to Questions

**Comments to the Author**

1. If the authors have adequately addressed your comments raised in a previous round of review and you feel that this manuscript is now acceptable for publication, you may indicate that here to bypass the “Comments to the Author” section, enter your conflict of interest statement in the “Confidential to Editor” section, and submit your "Accept" recommendation.

Reviewer #2: (No Response)

2. Is the manuscript technically sound, and do the data support the conclusions?

Reviewer #2: Yes

3. Has the statistical analysis been performed appropriately and rigorously? 

Reviewer #2: Yes

4. Have the authors made all data underlying the findings in their manuscript fully available?

Reviewer #2: Yes

5. Is the manuscript presented in an intelligible fashion and written in standard English?

Reviewer #2: Yes

6. Review Comments to the Author

Reviewer #2: The article provides information pertaining to the coagulation abnormalities in the local African context that could potentially assist with the treatment of patients with COVID and improve outcomes.

7. PLOS authors have the option to publish the peer review history of their article (what does this mean?). If published, this will include your full peer review and any attached files.

Reviewer #2: No

---

## [Author Response · Author response to Decision Letter 2]

5 May 2021

General Comments

The reviewer felt that the manuscript was substantially improved but there are still a number of grammatical and typographical errors that should be corrected. Again, it is recommended that the manuscript is reviewed by a native English language speaker.

Dear reviewers and editors

We have carefully consider your comments and we have incorporated all your comments and recommendations

The manuscript have also reviewed by a native English language speaker 

Specific comments

Comment 1: Abstract, background section: 

“Thus”, “taken as” and “COVID-19” was removed as per your suggestion in line 28 & 29

Comment 2: Abstract, method section: 

Insert trade mark: inserted

Remove tests and replace it with statistical analysis results: accepted and replaced

Comment 3: Result section: “with” is replaced with ‘in’ in line

Comment 4: abstract, conclusion section. 

Replace “around” with “more than”

Comment5: Introduction section: 

Dear editor and reviewers, all comments and suggestions given in introduction section were accepted and amended accordingly

Remove “heavily” in line 52

Replace 49,578, 490 with “more than 1.4 million” in line 53

Insert with lung injury in line 56

Remove and critical cases with

Comment 6: method section:

Dear editor and reviewers, all comments and suggestions given in method section were accepted and amended accordingly

Insert “personal identifying” in line 87: accepted and inserted

Delete “time”, “its” in line 107 and 108

Replace Platelet with platelets in line 108

Comment 6: Result section:

Dear editor and reviewers, all comments and suggestions given in result section were accepted and amended accordingly

Delete Y in line 126

Delete N=455 in table 1

Replace “higher” with “elevated”

Move heading of table 3 to the top

Comment 6: Discussion section:

Dear editor and reviewers, all comments and suggestions given in discussion section were accepted and amended accordingly

Replace “and” with “is” in line 153

Replace in “patients with COVID” with “and” in line 151

Delete “resulting in increased mortality and hospitalization and ICU admission” in line 153

Change manifest in to manifests and increase in to increases

Delete “(9.5%-47%)” in line 160

Insert “More frequently” in line 176 and 177

Rephrase the sentences from line 188-193

Before: The degree of coagulation abnormalities in critically ill non-COVID patients correlates with disease severity and predict the risk of thrombosis, the need for ventilator support, and mortality. Published studies support that COVID-19-associated coagulopathy is characterized by a decreased platelet count (9, 36-38). Patients with critical COVID-19 infection and a cytokine storm have an extreme hyper-coagulable state. 

Modified: Published studies indicate that COVID-19-associated coagulopathy is characterized by a decreased platelet count (9, 36-38)and a cytokine storm with an extreme hyper-coagulable state.

Comment 6: Discussion section:

Replace “APTT” with “clotting time assays”

Response: Dear editors and reviewers all of the above listed comments, recommendations and suggestions are accepted, replaced and corrected and we thank you very much for your valuable comments & time.

Looking forward to hearing from you. Thank you again for your consideration! 

Sincerely, 

Shambel Araya (BSc, MSc, PhD fellow)

---

## [Decision Letter · Decision Letter 3]

26 May 2021

Blood coagulation parameter abnormalities in hospitalized patients with confirmed COVID-19 in Ethiopia

PONE-D-20-35898R3

Dear Dr. Araya,

We’re pleased to inform you that your manuscript has been judged scientifically suitable for publication and will be formally accepted for publication once it meets all outstanding technical requirements.

Kind regards,

Elizabeth S. Mayne, M.D.

Academic Editor

PLOS ONE

Additional Editor Comments (optional):

Reviewers' comments:

Reviewer's Responses to Questions

**Comments to the Author**

1. If the authors have adequately addressed your comments raised in a previous round of review and you feel that this manuscript is now acceptable for publication, you may indicate that here to bypass the “Comments to the Author” section, enter your conflict of interest statement in the “Confidential to Editor” section, and submit your "Accept" recommendation.

Reviewer #2: All comments have been addressed

2. Is the manuscript technically sound, and do the data support the conclusions?

Reviewer #2: Yes

3. Has the statistical analysis been performed appropriately and rigorously? 

Reviewer #2: Yes

4. Have the authors made all data underlying the findings in their manuscript fully available?

Reviewer #2: Yes

5. Is the manuscript presented in an intelligible fashion and written in standard English?

Reviewer #2: Yes

6. Review Comments to the Author

Reviewer #2: The information contained in the article will add to much needed local African data pertaining to COVID-19 and assist in appropriate care of patients in order to improve clinical outcomes.

7. PLOS authors have the option to publish the peer review history of their article (what does this mean?). If published, this will include your full peer review and any attached files.

Reviewer #2: No

---

## [Editor Report · Acceptance letter]

28 May 2021

PONE-D-20-35898R3 

Blood coagulation parameter abnormalities in hospitalized patients with confirmed COVID-19 in Ethiopia 

Dear Dr. Araya:

I'm pleased to inform you that your manuscript has been deemed suitable for publication in PLOS ONE. Congratulations! Your manuscript is now with our production department. 

Kind regards, 

on behalf of

Dr. Elizabeth S. Mayne 

Academic Editor

PLOS ONE